# Domain loss enabled evolution of novel functions in the snake three-finger toxin gene superfamily

Ivan Koludarov [1,7] ✉, Tobias Senoner [1,7], Timothy N. W. Jackson[2], Daniel Dashevsky [3], Michael Heinzinger [1], Steven D. Aird[4] & Burkhard Rost[1,5,6]

Three-finger toxins (3FTXs) are a functionally diverse family of toxins, apparently unique to venoms of caenophidian snakes. Although the ancestral function of 3FTXs is antagonism of nicotinic acetylcholine receptors, redundancy conferred by the accumulation of duplicate genes has facilitated extensive neofunctionalization, such that derived members of the family interact with a range of targets. 3FTXs are members of the LY6/UPAR family, but their non-toxin ancestor remains unknown. Combining traditional phylogenetic approaches, manual synteny analysis, and machine learning techniques (including *AlphaFold2* and *ProtT5*), we have reconstructed a detailed evolutionary history of 3FTXs. We identify their immediate ancestor as a non-secretory LY6, unique to squamate reptiles, and propose that changes in molecular ecology resulting from loss of a membrane-anchoring domain and changes in gene expression, paved the way for the evolution of one of the most important families of snake toxins.

The low cost of accumulating sequence data has permitted molecular evolutionary studies to focus on the central role of nucleic acid sequences in "storing and transmitting" biological information. Furthermore, the linear nature (writable as 1D-strings of letters) of such sequences makes them readily amenable to computational analyses that increase our understanding of ways in which traits evolve and affect their functional roles in the life histories of the organisms that possess them. However, as functional traits, including those of individual molecules, are relational, we cannot understand their evolution by focusing on sequences alone[1,2]. Gene-products act as three-dimensional (3D) protein structures therefore one-dimensional (1D) gene/protein sequence data alone are insufficient to explain molecular mechanisms or evolutionary history. Nucleic acids "store" information, deployment of which is selective and context-specific[3]. A multi-level approach is therefore required to understand

the phenotypic consequences of sequence variation upon which selection acts.

Venom systems are excellent to study the impact of gene sequence change upon protein function because most proteinaceous toxins are encoded by single genes and adapted for specific functions when injected into target organisms[4]. Few other systems offer such a clear causal pathway from genetics to ecology. This tractability helped investigations of toxin-derived molecules as drugs and their utilization as investigational ligands[5]. Here, we chronicle the origins and innovations of an ancient gene superfamily that includes one of the most widespread and functionally diverse toxin families in snake venoms: three-finger toxins (3FTXs) derived from a superfamily called lymphocyte antigen 6 (LY6) or urokinase-type plasminogen activator receptors (UPAR)[6]. Members of this superfamily are located on several chromosomes in vertebrates and are also present in many groups of

[1]TUM (Technical University of Munich) Department of Informatics, Bioinformatics & Computational Biology—i12, Boltzmannstr. 3, 85748 Garching/Munich, Germany. [2]Australian Venom Research Unit, Department of Biochemistry and Pharmacology, University of Melbourne, Melbourne, VIC, Australia. [3]Australian National Insect Collection, Commonwealth Scientific & Industrial Research Organisation, Canberra, ACT, Australia. [4]7744-23 Hotaka Ariake, 399-8301 Azumino-shi, Nagano-ken, Japan. [5]Institute for Advanced Study (TUM-IAS), Lichtenbergstr. 2a, 85748 Garching/Munich, Germany. [6]TUM School of Life Sciences Weihenstephan (WZW), Alte Akademie 8, Freising, Germany. [7]These authors contributed equally: Ivan Koludarov, Tobias Senoner. ✉ e-mail: koludarov@rostlab.org

invertebrates, indicating their ancient origin. Although 3FTXs appeared relatively recently in this superfamily, they are its best characterized members, and as a result, the LY6/UPAR superfamily is also sometimes referred to as "toxin-like proteins" (TOLIP), or the snake-toxin-like superfamily[7].

LY6/UPAR proteins possess a characteristic "LU" or "3FF" ("three-finger fold"; Fig. 1). The group includes membrane-bound proteins such as multimeric UPARs, which have a C-terminus that is post-translationally removed to attach a glycophosphatidylinositol (GPI) anchor (MaD: the "membrane-anchoring domain/region"), and

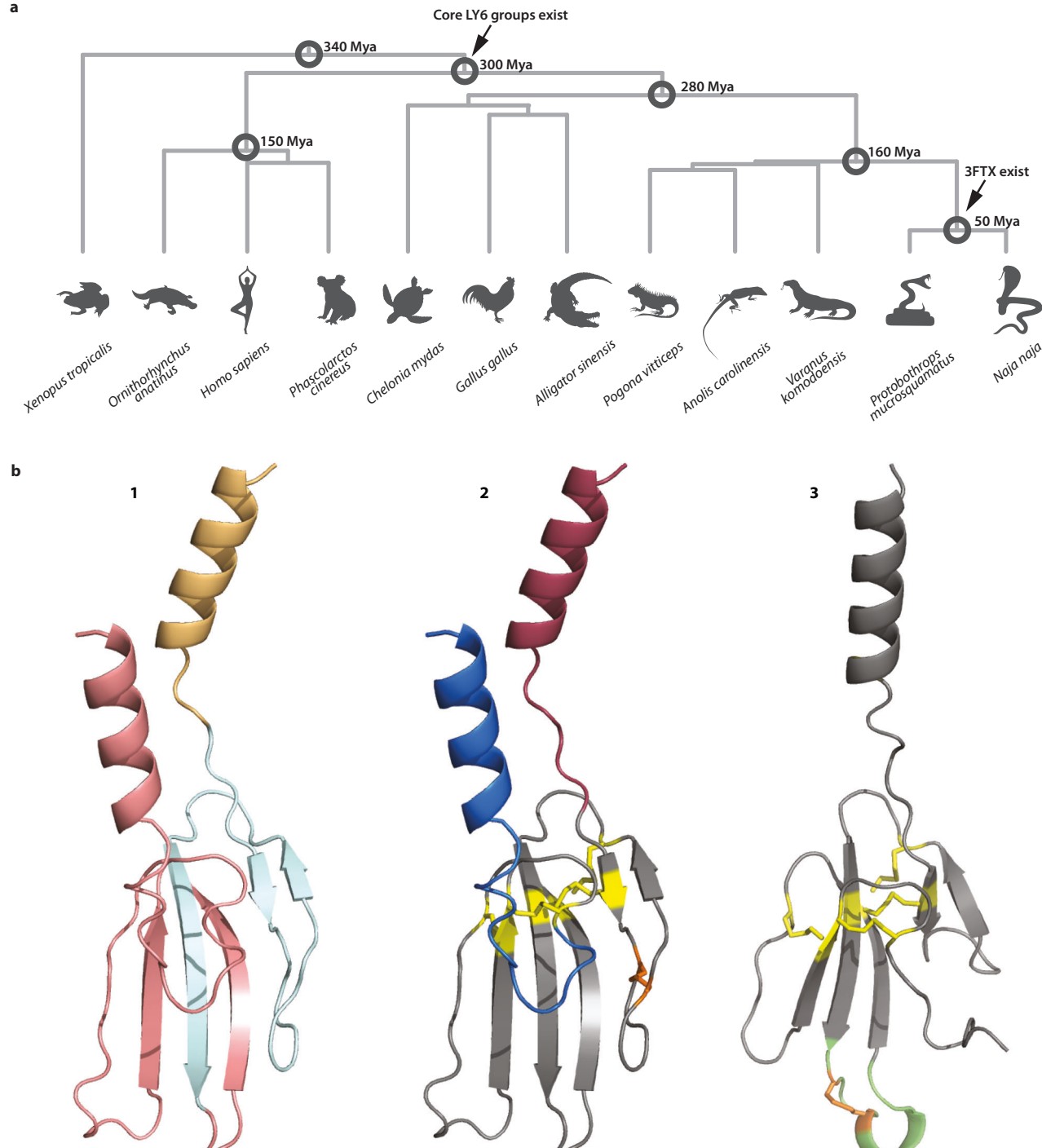

**Fig. 1 | Primary species and proteins investigated in this study. a** Species and their phylogenies on which this study is centered, including estimated divergence times in millions of years ago; (**b**) Representative structural diversity in 3FTX/LY6 family. b1 Structure of a typical membrane bound LY6 protein (human LYNX1), color highlights parts of the molecule that are encoded by different exons: exon one in orange, exon two in light blue, exon three in pink. b2 The same structure with key elements highlighted: signal peptide in maroon, membrane-anchoring domain in blue, cysteine bonds common to the entire 3FTX/LY6 family in yellow, cysteine bond that disappears in derived 3FTXs in orange. b3 Structure of a derived 3FTX (alpha-bungarotoxin), highlighting changes in structure from a plesiotypic form: cysteine bond new to long-chain 3FTXs in orange, extended loop in green, common cysteine bonds in yellow, notice missing cysteine and membrane-anchoring domain (cf. 2). All structures computed as a part of this study (see "Methods" and Supplementary Data 2).

**Table 1 | Existing knowledge of human LY6 proteins, homologs of 3FTXs**

| Protein name | Sketch of known information |
|---|---|
| GPIHBP1 Glycosylphosphatidylinositol anchored high density lipoprotein binding protein 1 | Capillary endothelial cell membrane protein assisting lipoprotein lipase in processing dietary lipoprotein. |
| LY6H Lymphocyte Antigen 6 Family Member H | Membrane-anchored; assumed to modulate nAChRs activity; seems to inhibit alpha-7/CHRNA7 signaling in hippocampal neurons. |
| LY6L Lymphocyte Antigen 6 Family Member L | Membrane-anchored. |
| LY6E Lymphocyte Antigen 6 Family Member E | Anchored cell surface protein regulating T-lymphocytes proliferation, differentiation, and activation. |
| LY6D Lymphocyte Antigen 6 Family Member D | Predicted as membrane-anchored; predicted to be involved in lymphocyte differentiation. |
| LYNX1 LY6/Neurotoxin 1 | Membrane-anchored; modulates functional properties of nAChRs to prevent excessive excitation, and hence neurodegeneration. |
| SLURP2 Secreted LY6/UPAR Related Protein 2 | Secreted; binds and may modulate the functional properties of nAChRs; may regulate keratinocytes proliferation, differentiation and apoptosis. |
| LYPD2 LY6/PLAUR Domain-Containing Protein 2 | Predicted to be located in the extracellular region and plasma membrane. |
| SLURP1 Secreted LY6/UPAR Related Protein 1 | Secreted; found to be a marker of late differentiation of the skin; may be involved in the regulation of intracellular Ca(2+) signaling in T cells. |
| LY6K Lymphocyte Antigen 6 Family Member K | Predicted in the cell surface, cytoplasm, and plasma membrane; predicted to be active in acrosomal vesicle and to be involved in binding of sperm to zona pellucida. |
| PSCA Prostate Stem Cell Antigen | Membrane-anchored; highly expressed in prostate, also expressed in bladder, placenta, colon, kidney, and stomach; possibly involved in regulation of cell proliferation; inhibits nicotine-induced signaling, in vitro. |

Based on UniProt and Entrez databases. In order of chromosomal position downstream from *TOP1MT* gene.

secretory proteins lacking this region (for example secreted LY6/UPAR-related protein SLURP and 3FTXs)[6–8]. Many LY6/UPAR proteins function as lymphocyte antigens. Others bind and modulate nicotinic acetylcholine receptors (nAChRs, Table 1)[7]. The latter function is mediated by the cysteine-rich 3FF region, typically involving 60–74 residues. While this region has no enzymatic activity, 3FTXs have co-opted their affinity for the nAChR as the primordial toxic function[9]. In snakes, these genes were recruited to serve as neurotoxins, leading to an explosive diversification of toxin sequences and activities[7,10]. Plesiotypic forms of the 3FTX family possess 10 cysteines in 5 disulfide bonds (Fig. 1). These forms exert "α-neurotoxicity" by antagonizing nAChRs, with some forms exhibiting greater affinity for reptile or bird, rather than mammalian receptors[8]. This primordial theme has been extensively varied within the venom systems of colubroid snakes (particularly members of the family *Elapidae*), generating considerable evolutionary and ecological impact[9,10].

The relation between plesiotypic toxins and derived elapid forms remains difficult to discern despite substantial research on 3FTXs. Molecular phylogenies of 3FTXs are typically restricted to either plesiotypic or derived forms, either to narrow the research question or because Bayesian phylogenies including both types almost invariably collapse into extensive, uninformative polytomies[11,12]. Ancestral relations within the LY6/UPAR superfamily remain unclear, although they have long been suspected to be orthologs of *LYNX1* and its homologs on Chromosome 8 in humans[8].

While there has been an explosion of high-quality sequence data since the 1960s[13] many sequences remain unannotated and inferring structure or activity from sequence has been challenging. Recent advances in artificial intelligence (AI) promise to mitigate these bottlenecks. As of spring, 2023, AlphaFold2[14] has been used to make accurate predictions of 3D protein structure for over 200 million proteins[15]. This wealth of data allows us to infer similarities between proteins based on their structures, such as FoldSeek[16]. Independently, another breakthrough built upon advances in natural language processing (NLP) by analogizing words with amino acids and sentences with proteins. This simple adaption enabled the transfer of NLP concepts to biology by learning some aspects of the "language" of life as written into protein sequences through AI models dubbed protein Language Models (pLMs)[17–19].

In this study, we combined reliable 3D structure predictions from AlphaFold2[14] and embeddings from the pLM ProtT5[18] with traditional bioinformatic analyses and extensive manual microsyntenic analyses. Using public sequence data (including newly annotated and re-annotated genes), we examined the *LY6* genomic region in various vertebrate genomes to reconstruct the evolutionary history of 3FTXs and non-toxin family members. For simplicity, we refer to genes/proteins in this region as "LY6", while using "TOLIP" to refer to the larger parental group of UPARs. Our approach[20–23] enabled us to reconstruct the evolutionary history of the *LY6* gene family with a high level of detail and to catalog the emergence of novel structures and functions that led from an ancestral LY6 to the radiation of deadly snake venom 3FTXs.

## Results

### Genomic analysis reveals an evolutionary stable LY6 cluster
To distinguish distantly related *TOLIP UPAR/PLAUR* genes from *LY6s* that are direct homologs of snake *3FTXs*, we surveyed genomes of 15 species of *Tetrapoda* (four-limbed vertebrates, see Table 2 in "Materials and Methods") for regions syntenic with snake *3FTXs* (Fig. 2). We identified the mitochondrial topoisomerase, *TOP1MT*, as the best marker gene for this region. In all genomes surveyed, *TOP1MT* was present in a single copy and either flanked the *3FTX/LY6* genomic region or was in the middle of it (Supplementary Fig. 1). Another useful marker gene is *THEM6* (Thioesterase Superfamily Member 6 − UniProt ID Q8WUY1 for a human form) present in mammals, some reptiles (including snakes), and *Xenopus* (clawed frog). This gene is located near *SLURP1* homologs, implying that its stable synteny is rather ancient.

We located all *LY6/3FTX* genes (both full-length and missing exons) using previously established methods of BLASTing exons across a genomic region[20–23]. After this, we described multiple genes, pseudogenes, and orphan exons in each species examined. In most cases, the *LY6/3FTX* cluster was located downstream from *TOP1MT*. It spanned several megabase pairs (Mbp), contained 8–35 isoforms (10, on average, for non-venomous species) from the *LY6/3FTX* family and rarely included genes from any other protein family (Fig. 2, Supplementary Fig. 1). These syntenic arrangements were extremely conserved among all species sampled, with the notable exception of

**Table 2 | Genomes used**

| Species | ID | Genome URL |
|---|---|---|
| *Alligator sinensis* | GCA_000455745.1 | https://www.ncbi.nlm.nih.gov/genome/?term=GCA_000455745.1 |
| *Anolis carolinensis* | GCA_000090745.2 | https://www.ncbi.nlm.nih.gov/genome/?term=GCA_000090745.2 |
| *Bungarus multicinctus* | CNP0002662 | https://db.cngb.org/search/project/CNP0002662/ |
| *Chelonia mydas* | GCA_015237465.2 | https://www.ncbi.nlm.nih.gov/genome/?term=GCA_015237465.2 |
| *Gallus gallus* | GCA_000002315.5 | https://www.ncbi.nlm.nih.gov/genome/?term=GCA_000002315.5 |
| *Homo sapiens* | GCA_000001405.29 | https://www.ncbi.nlm.nih.gov/genome/?term=GCA_000001405.29 |
| *Naja naja* | GCA_009733165.1 | https://www.ncbi.nlm.nih.gov/genome/?term=GCA_009733165.1 |
| *Ornithorhynchus anatinus* | GCA_004115215.4 | https://www.ncbi.nlm.nih.gov/genome/?term=GCA_004115215.4 |
| *Phascolarctos cinereus* | GCA_002099425.1 | https://www.ncbi.nlm.nih.gov/genome/?term=GCA_002099425.1 |
| *Pogona vitticeps* | GCA_900067755.1 | https://www.ncbi.nlm.nih.gov/genome/?term=GCA_900067755.1 |
| *Protobothrops mucrosquamatus* | GCA_001527695.3 | https://www.ncbi.nlm.nih.gov/genome/?term=GCA_001527695.3 |
| *Pseudonaja textilis* | GCA_900518735.1 | https://www.ncbi.nlm.nih.gov/genome/?term=GCA_900518735.1 |
| *Python bivittatus* | GCA_000186305.2 | https://www.ncbi.nlm.nih.gov/genome/?term=GCA_000186305.2 |
| *Varanus komodoensis* | GCA_004798865.1 | https://www.ncbi.nlm.nih.gov/genome/?term=GCA_004798865.1 |
| *Xenopus tropicalis* | GCA_000004195.4 | https://www.ncbi.nlm.nih.gov/genome/?term=GCA_000004195.4 |

snakes (Fig. 3), and most genes were shared by all members within each taxonomical class, i.e., Reptilia or Mammalia.

The *3FTX* cluster is dramatically expanded in the Indian cobra (*Naja naja*) one of the very few elapid genomes sequenced to-date at the chromosomal level[23]. This pattern conformed to the central role of this toxin family in the evolution of venoms in elapid snakes[9]. Another unique feature of elapid genomes was the presence of two genes in the Indian cobra located outside the *TOP1MT* neighborhood (on Chromosome 3). Those were single-copy *LY6/3FTX* family genes on Chromosomes 1 and 4. We managed to locate an ortholog for this gene only on Chromosome 4 in the Eastern brown snake (*Pseudonaja textilis*) genome, whereas both were found in the many-banded krait (*Bungarus multicinctus*). We found no related genes (or stray exons) in homologous genomic regions of any other organism analyzed. Many *3FTX* genes have a 4-exon structure distinct from the typical 3-exon structure (Fig. 3). All additional exons are duplicated and pseudogenized copies of exon 3, inserted before the original exon 3. We did not find evidence of protein production involving these redundant exons. One of the genes in Bearded dragons (*Pogona*) likely has a similar exonic structure (Povi_7 in our dataset, see Supplementary Fig. 1); however, we could only detect pieces (encoding 4–5 aa each) of the duplicated exon, not its full, even if pseudogenized, sequence. A human *LY6K* gene possesses a very similar feature, with its third exon duplicated twice, and all three isoforms apparently functional. Lastly, human *LYPD2-SLURP1* has a known "read-through" 4-exon chimera (exons 1 and 2 from *LYPD2* concatenated with exons 2 and 3 of *SLURP1*) that is very likely a uniquely mammalian feature, given that in all other genomes surveyed those genes have a different orientation and are facing each other. However, genomic annotations for both *Pogona* and *Varanus* predict similar read-throughs involving *LYPD2* and one of the unique reptilian *LY6* genes located upstream from *LYPD2*. Whether this is real or an annotation artifact due to the projection of human genomic annotation onto lizard genomes (in all the cases, the read-through occurs right before *THEM6* gene) remains an open question.

**Protein 3D-structural analysis**

We used the ColabFold[24] implementation of AlphaFold2[14] to predict full sequence (where available) and mature protein 3D structures for all 1,427 LY6 family proteins in our dataset (Supplementary Data 1). Where experimental 3D structures were available, we validated the Alpha-Fold2/ColabFold predictions against them (see Supplementary Data 1 for comparison and Supplementary Data 2 for structures). All proteins

in the dataset were observed and predicted in the same canonical 3FTX-domain architecture. They differed in their compactness (density of inter-residue contacts) and loop lengths (continuous regions with neither helix nor strand). Most LY6 genes (~86%) possessed a lengthy C-terminal "tail", in some cases predicted to adopt an alpha helix. In humans that extension facilitates GPI-assisted membrane anchoring[6,7]. Following the broad classification of 3FTX accepted in literature[10–12], we classified all toxins in our dataset into 4 categories: plesiotypic (with ancestral LY6 cysteine arrangement and no loop extensions), short chain (8 cysteines), long chain (10 cysteines, of which one pair is novel, loop extension present) and non-standard (that do not fit into any of the previous categories; also called "non-canonical" in literature).

**Phylogenetics (ExaBayes, iqTree, DALI)**

We removed signal peptides from sequences (to avoid using potentially mis-predicted N-termini) and aligned cut sequences with MAFFT[25]. The resulting alignments seeded the Bayesian (ExaBayes[26]) and Maximum Likelihood (iqTree[27]) phylogenetic trees. We also generated a phylogenetic tree based on DALI (Distance-matrix ALIgnment[28]) comparisons of predicted 3D structures. While smaller protein clades were consistent across the three trees, their inter-relationships differed between the three trees from ExaBayes, iqTree, and DALI (Fig. 4, Supplementary Fig. 2). The Bayesian ExaBayes phylogeny recovered a single monophyletic 3FTX clade albeit one that was plagued by extensive polytomy, but still more robust than any previously published phylogeny (Supplementary Fig. 2c). This clade was sister to LY6E-like sequences, including a clade of toxicoferan LY6 to which we refer as "pre-3FTX" (Fig. 2). The iqTree phylogeny looked largely similar, but suffered from low bootstrap values for most nodes, except those that defined some of the smaller clades. In contrast, the DALI phylogeny recovered the vast majority of 3FTXs in two clades, one with primarily cat-eyed snake (*Boiga*) 3FTXs and other colubrid toxins, and the second including plesiotypic toxins from other snake families and more derived elapid toxins. 3FTXs were even more polyphyletic in the DALI than the iqTree topology, because several 3FTXs were orphaned on single branches, sisters to a clade containing the vast majority of LY6 sequences and the two main 3FTX clades. Some findings were consistent across phylogenies. Secretory LY6 proteins lacking the MaD were distributed among various clades, indicating that this domain has been lost convergently on multiple occasions, and non-standard 3FTXs whose cysteine patterns deviate from the stereotypical major toxin groups are found throughout the gene family

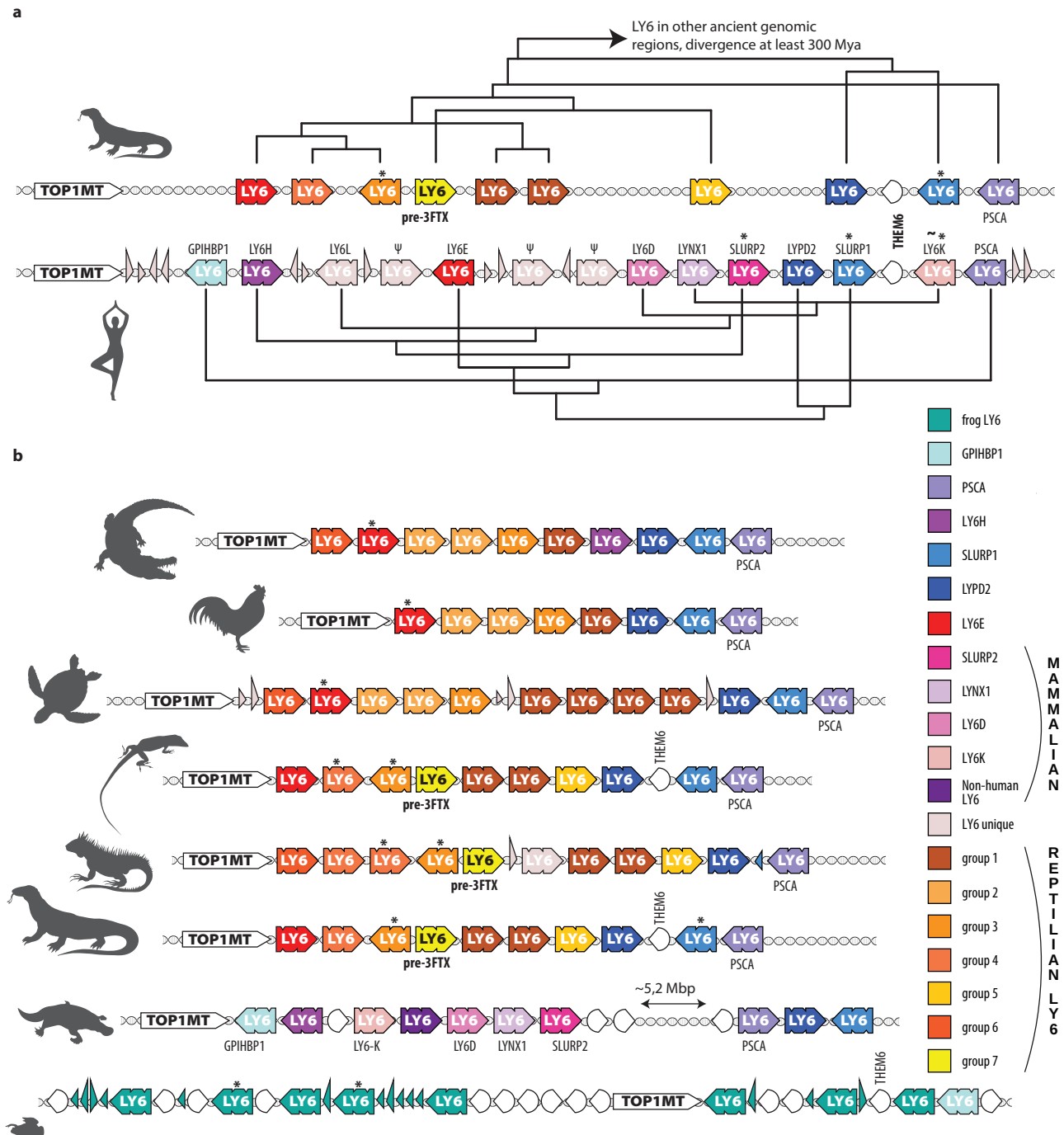

**Fig. 2 | Syntenic schema derived from combined phylogenetic and genomic analyses. a** Compares *LY6/3FTX* clusters from Komodo dragons and humans. Gene names are from annotations. Unless marked with Ψ (pseudogene), gene relationships are based on our phylogenetic analyses. **b** Provides a syntenic map for representatives of tetrapod clades. *LY6* genes are colored according to the legend, with *LY6 unique* standing for genes that do not have direct orthologs in surveyed species. *TOP1MT* are white, and other genes are depicted as hollow semi-circular, white arrowheads, with narrow triangles representing orphan exons. Genes encoding forms predicted to be secreted are indicated with asterisks (*). Human *LY6K* is marked with (~*) because only some forms are secreted (see text for details).

phylogeny and across taxa (see Supplementary Fig. 2 for details). Both iqTree and ExaBayes phylogenies (and DALI, to a lesser extent) revealed several LY6 clades that were unique to reptiles. Those clades do not have any names, unlike mammalian homologs of human genes, and as far as we can tell we are the first to characterize them as groups. We labeled them "reptilian LY6 groups" with numbers 1 to 7 (Fig. 4). For the rest of the study we rely on Bayesian phylogeny, which seems to offer the closest representation of reality.

## Embedding maps

We used the pLM ProtT5 to generate multidimensional (1024D) vector representations (embeddings) from single protein sequences. Essentially, these vectors resemble the simplicity of 20D vectors obtained by computing amino acid composition, i.e. the fraction of each of the 20 amino acids in a protein. However, while amino acid composition vectors ignore interactions between pairs of residues, pLM embeddings reflect such effects, on average. We computed embedding

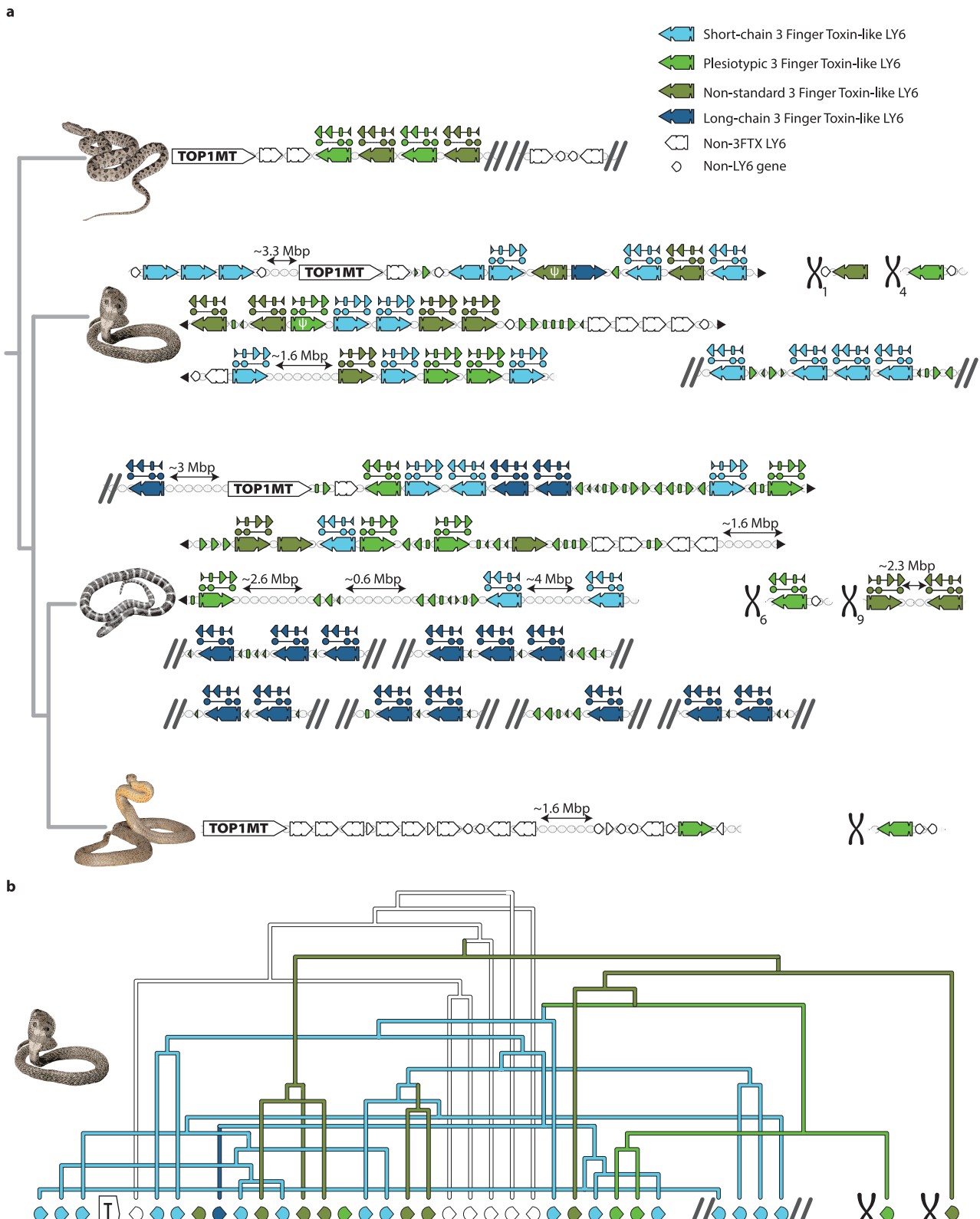

**Fig. 3 | Snake *3FTXs*. a** *3FTX* genomic region in snakes, top to bottom: *Protobothrops muscrusquamatus*, *Naja naja*, *Bungarus multicinctus* and *Pseudonaja textilis*. Non-3FTX *LY6* in white, genes unrelated to the *LY6/3FTX* family indicated by white semi-circular arrowheads, short-chain *3FTX* genes in light blue, long-chain *3FTX* genes in dark blue, plesiotypic *3FTX* genes in bright green and non-standard *3FTX* genes in moss green. Exonic shapes above genes show non-canonical exonic structure with duplicated Exon 3. In all cases, only a single copy of exon 3 is used for a mature sequence, while another is pseudogenized. *TOP1MT* gene indicates continuous genomic region that is homologous to *LY6* cluster region shown in Fig. 2. Other chromosomal regions are indicated by an X with numbers where the exact chromosome is known. Ends of bioinformatic scaffolds are indicated with two diagonal parallel lines, where genomic regions continue to a line below is indicated with black triangles. Orphan exons are indicated as follows: large triangle for exon 3, rounded small rectangle for exon 2, and small triangle for exon 1. **b** Phylogenetic relationships between cobra genes and their respective genomic position, pseudogenized genes are marked with Greek letter psi.

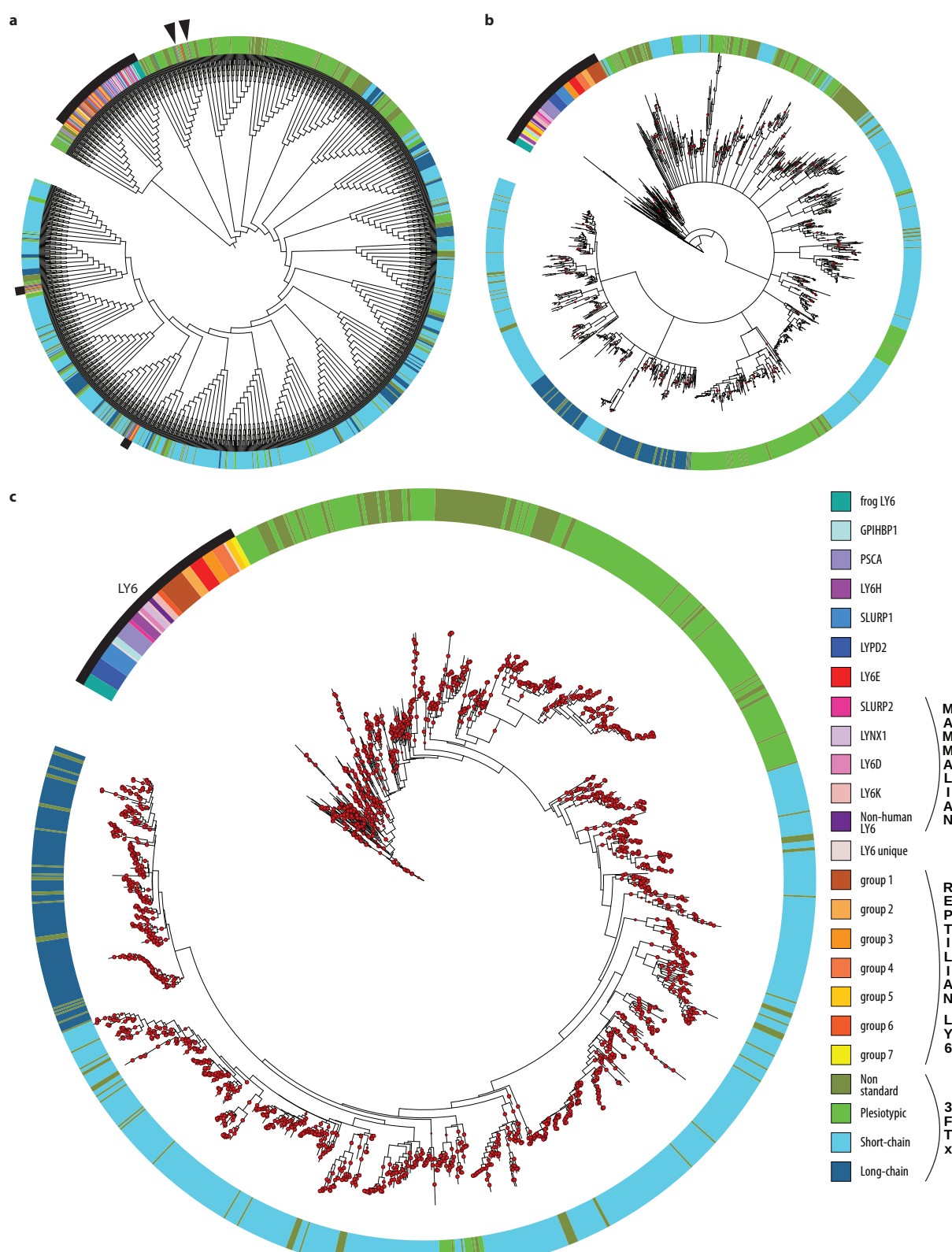

**Fig. 4 | Phylogenetic trees for the non-redundant dataset used in this study.** Leaves are colored according to the figure legend. LY6s are indicated by a thick black arc and black arrows. **a** DALI branch lengths have no meaning due to the nature of the analysis. **b** Maximum-likelihood nodes with bootstrap higher than 85% are marked by red dots. **c** Bayesian inference: Node support higher than 85% is indicated by a red dot. See Supplementary Materials 3 for full resolution trees.

vectors for all proteins in two sets: the core LY6/3FTX dataset (1427 sequences), and the entire InterPro[29] TOLIP family (2466 non-identical sequences on May 1st, 2023, of which 163 were already part of our dataset). To facilitate visual inspection, we projected the 1024D

embeddings onto 3D through UMAP[30] (Fig. 5, https://rostspace.onrender.com/ for an interactive plot). The projected embedding space distribution for LY6s from the *TOP1MT* region (including 3FTXs) was similar between the two data sets (core LY6/3FTX vs. TOLIP in

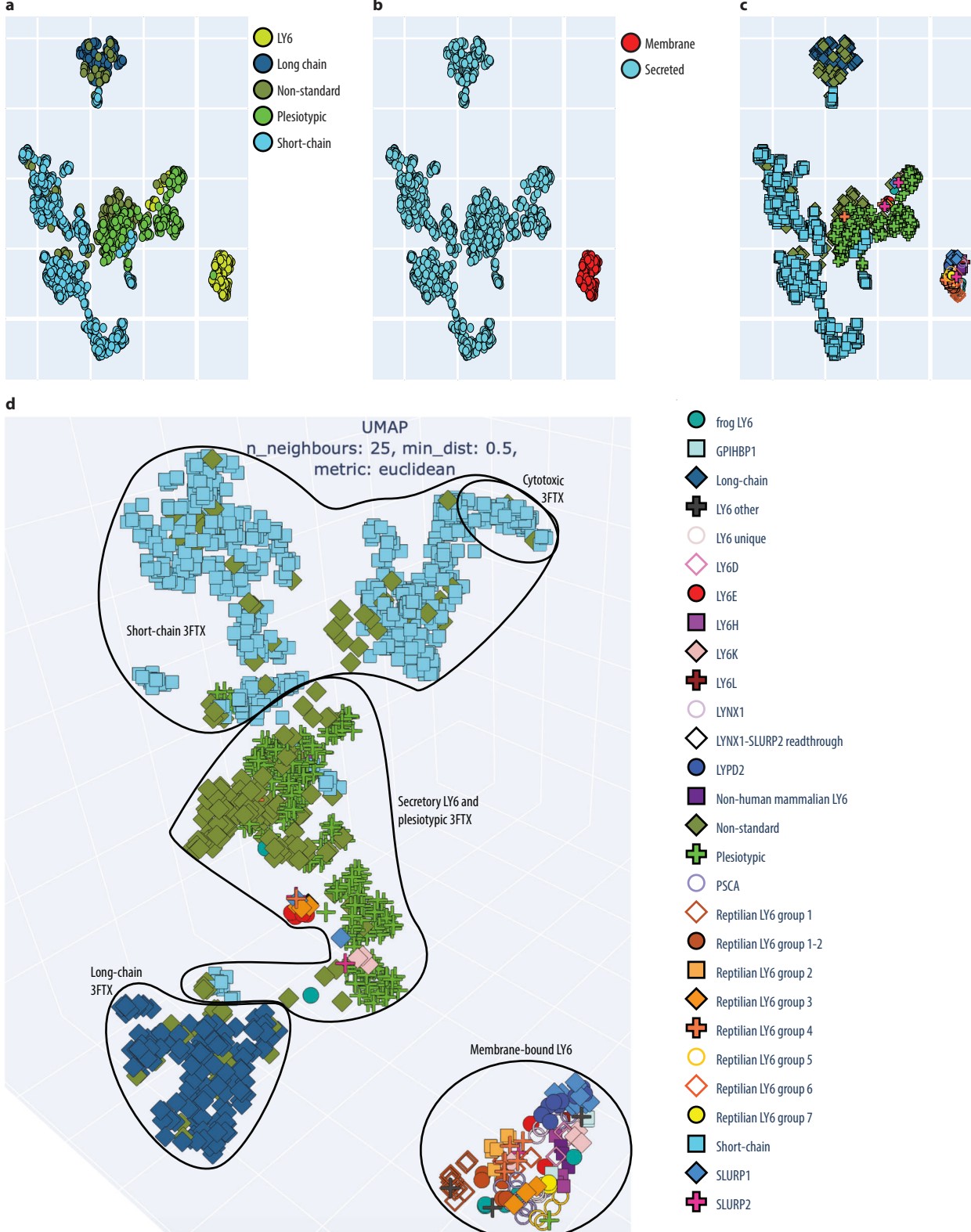

**Fig. 5 | Rendering of 3D UMAP projection embedding space for LY6/3FTXs.** Key structure-function clusters are circled and named. Each protein in our set was represented by a 1024-dimensional vector of the ProtT5 pLM embeddings[18]. These 1024D vectors were projected by UMAP[30] onto 2D (Panels a through c) and 3D (Panel d). **a** Shows the distribution of subgroups within the family. **b** Shows predictions of secreted and membrane-bound proteins. **c** and **d** are different representations of the same information on smaller clades within the family, thus, only a single figure legend is given. The resulting embedding map seeded hypotheses on the hypothetical evolutionary trajectory of this family in protein space that were compared to existing sequence- or structure-based approaches. Clustering, especially with the 3D projection, obscures many points and skews the actual distance. Compare it with sequence-based clustering in Supplementary Fig. 3 and TOLIP clustering in Supplementary Data 3. The full detail of interactive plots available at: https://rostspace.onrender.com/. Embeddings are available in Supplementary Data 4.

InterPro). All LY6 with a MaD occupied a compact area of the projected sequence space. Secretory LY6s and 3FTXs formed a separate, more diffuse supercluster. Plesiotypic 3FTXs formed a distinct group along with LY6s lacking the MaD domain, e.g., SLURPs. Toxins of colubrid snakes were found along the length of this elongated cluster while plesiotypic toxins of elapid snakes were located together at one end (Fig. 5 and interactive plot). Near this end, we found another long cluster that included all short-chain derived 3FTXs. Once again, relations within this cluster appeared relevant: cytotoxic 3FTXs were gathered at the end of one extended cluster whereas another small subcluster contained most toxins from hydrophiine snakes. There were also three outliers in this cluster: two sequences from *Helicops leopardinus* (leopard keelback) which have convergently lost the same disulfide bond as elapid short-chain toxins and one sequence from *Causus rhombeatus* (night adder) that appears to be a misattributed elapid sequence, possibly due to cross-contamination between sequencing samples. Long-chain toxins formed their own distinct cluster, which is far more compact than those of plesiotypic or short-chain 3FTXs.

## Discussion

### Embeddings boost understanding of origins of three-finger toxins

Our results illustrate why reconstructing the early evolution of 3FTXs has proven difficult[8]. The two traditional phylogenetic methods compared, provided conflicting results. The Bayesian phylogeny exhibited unresolved polytomies similar to those that have plagued previous efforts to incorporate plesiotypic and derived toxin sequences into the same trees (Fig. 4 and Supplementary Fig. 2c). Despite the polytomous substructure of the 3FTX clade in the Bayesian phylogeny, our synteny- and pLM-based methods are concordant with hypothesizing that the whole toxin clade had a single origin, rather than multiple origins, as implied by the structure-based DALI phylogeny.

Previous approaches relied mostly on statistical analysis of related proteins via multiple sequence alignments (MSAs). In contrast, the main idea behind pLMs is to let an artificial neural network (ANN) learn how to "read" a protein sequence, i.e., how to make an amino acid sequence computer-readable. Similar to a child learning a new language, this is achieved by training *transformers* (the underlying ANN architecture of the most successful LMs and pLMs) on filling out cloze tests for LMs or residues for proteins. While repeating this task on billions of sentences or protein sequences, the (p)LM learns to detect re-occurring patterns by encoding them in its internal, trainable weights. This is achieved via the *attention mechanism*[31] at the core of each transformer, which learns for each input token (a word or an amino acid) a weighted average over all the other tokens in the same sentence or protein sequence. The crucial advantage of training the network on cloze tests is that we can directly leverage information from the gigantic amount of unlabeled data, i.e., we need only sequential data without any other information on the language-specific grammar. The knowledge acquired by pLMs during this process can later be transferred to any other task by providing a sentence/sequence as input to the model and extracting its *hidden states* (internal activations of the ANN). This is often dubbed *transfer learning*. Akin to a new experimental technique, the numerical vectors, *embeddings*, extracted this way provide a novel perspective on the relationship between proteins, orthogonal to established approaches such as homology-based inference. Contrary to most existing approaches which rely on MSAs, pLMs only need single protein sequences as input to provide protein comparisons and predictions[32–34]. These embeddings form a sequence space that reveals previously obscured biological information[35], an invaluable feature for studying gene evolution and the approach at the center of the present study. Despite their success, the complexity of large transformers makes it hard to interpret why a specific input sequence leads to a

specific embedding. An AI subfield, *XAI* (explainable artificial intelligence), is emerging to address such questions[36,37], hopefully at some future point[38].

Our pLM ProtT5 embedding model differentiated the major molecular and functional forms of 3FTXs that have been identified by decades of structure-function research. Impressively, endophysiological LY6, plesiotypic, short-chain, and long-chain toxins all form distinct clusters; toxins of shared activity, such as cytotoxins, even group together within these clusters (Fig. 5). Importantly, this grouping emerged intrinsically without requiring any expertise other than inputting the 3FTX/LY6 dataset or the entire InterPro[29] TOLIP family. Thus, using embeddings from pLMs clearly advances predictions of molecular activity from sequence data by analogy with prior knowledge and enables the annotation of uncharacterized proteins extracted from the genomic data with greater confidence than is possible using only phylogenies.

### 3FTXs evolved from a single-copy gene unique to toxicoferan reptiles

Identifying a common, endophysiological ancestor of exophysiological 3FTXs has proven challenging, although previous studies have suggested monomeric neuromodulatory LY6s, such as LYNXs and SLURPs, to be the most likely candidates[8]. Our embedding results put LYNXs in the main cluster of LY6 sequences while SLURPs and non-standard LY6Ks clustered with plesiotypic 3FTXs as the closest human homologs. This appears intuitive, as secretory SLURPs lack the MaD (membrane-anchoring domain) similar to 3FTXs; however, no LY6 group present in mammals was close to 3FTXs in our analyses. Instead, several unique reptilian groups of genes were indicated as possible homologs to the ancestor of 3FTXs. These clades of genes have no previous names assigned to them, so we labeled them (and other newly discovered groups) "*reptilian LY6 groups*" 1 to 7. Clustering methods applied to the embedding space identified reptilian LY6 group 3, and reptilian LY6 group 4 as the sequences closest in the embedding space to the 3FTXs. Interestingly, most members of this cluster possess a signal peptide that begins with MKT, a character that is rare in the LY6, but is by far the most common start in 3FTXs. Given other lines of evidence, this could be the result of molecular convergence. LY6 with a MaD cluster together, separated from those without, including these potential homologs to the ancestor of 3FTXs (groups 3, 4, 5 and 7), which are in turn much closer to 3FTXs. Our phylogenetic and syntenic analyses strongly suggest that MaDs have been lost repeatedly and convergently, which would imply that the embedding space captures structural features of the protein sequences rather than their phylogeny. The case for convergence is bolstered by the hierarchical clustering that places the candidates closer to other non-toxicoferan LY6 including secreted human forms than to the 3FTXs (Supplementary Fig. 4). Our genomic and syntenic results offer yet-another possibility: while members of *reptilian LY6 groups 3, 4* and *5* have been reconstructed from the genomes of snakes bearing 3FTX, *reptilian LY6 group 7* is only known from genomes of non-snake toxicoferans where it occurs in a similar microregion to where *3FTXs* are found in snake genomes (Fig. 6). Signal peptides of these reptilian group 7 genes begin with MK, but not MKT, and all possess a MaD. Our Bayesian phylogeny revealed this group (reptilian LY6 group 7) as a sister group to 3FTXs. Based on the above, we labeled it "pre-3FTXs."

Our syntenic analyses indicate that all reptiles have a genomic cluster (of varying copy number) of *LY6-like* genes between *TOP1MT* and the more conserved *SLURP1-like UPAR* genes. Given the stability and age of these syntenic groups, it is likely that each protein encoded by these genes possesses a conserved endophysiological function, including, in some cases, regulation of acetylcholinergic pathways. One of these copies in the cluster became the "*pre-3FTXs*" in the *Toxicofera*. After losing the membrane-anchoring region (convergently

**Fig. 6 | Synteny map of *LY6/3FTXs* in tetrapods and an inferred evolutionary scenario.** *LY6/3FTX* family genes are colored according to the legend. Genes unrelated to the *LY6/3FTX* family have been left out (except for *TOP1MT*, indicated by a white arrow with a T in it). Double diagonal bold lines signify breaks in bioinformatic scaffolds. Black triangular arrows show a continuation of the chromosomal region, compressed for the sake of space. The black chromosome icon specifies chromosomes different from those that house the *TOP1MT* region. Dotted outlines show pseudogenes and orphan exons.

with other secretory LY6s), and gaining increased affinity for nAChRs, this form gave rise to snake venom 3FTXs (Fig. 7). This led to a dramatic expansion of the cluster eventually resulting in genomes such as Indian cobra (*Naja naja*) with more than 30 *3FTX* genes, many orphan exons and pseudogenes that testify to a long history of "birth-and-death" evolution (Fig. 3, Supplementary Fig. 1). Likewise, the taxonomic distribution of leaves and branches of LY6/3FTX protein phylogenies clearly indicates constant emergence of new forms and copies in the genomic cluster.

Genomic remnants discovered in our study are consistent with past research, which has repeatedly found that *3FTX* genes are subject to high levels of diversifying selection and extremely high rates of molecular evolution[9,39,40]. Such rapid rates of evolution have been attributed partially to the relaxation of selective constraints on individual members of multigene arrays possessing a degree of functional redundancy[40]. Accumulation of *3FTX* genes in the Indian cobra (*Naja naja*) and many-banded krait (*Bungarus multicinctus*) genomes[23,41] provides a vivid example of this process. Unique syntenic arrangements of snake *3FTX* and *LY6* genes also suggest that some initial "*LY6E-like*" genes were lost to recombination with newly evolved *3FTXs*. If so, this would indicate possible "reverse-recruitment"[42], or

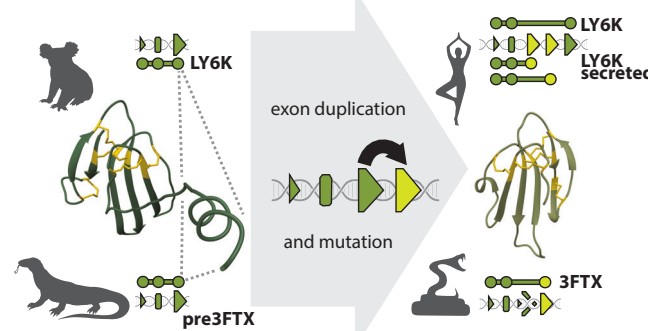

**Fig. 7 | Proposed mechanism of parallel evolution of membrane-anchored forms into secreted forms of *LY6* genes.** Small triangles stand for exon 1, rounded rectangles for exon 2, and large triangles for exon 3. Exons that make up a gene are represented by circles underneath genomic schemas and are connected via a line. Newly evolved exons are colored yellow. A pseudogenized exon is indicated by a shattered triangle.

"moonlighting" (adaptation of separate function)[43], in which 3FTXs, derived for envenomation functions, acquired regulatory functions of their endophysiological ancestors. Although details of such molecular ecology are beyond the scope of this paper, expression of "3FTX-like" proteins in diverse tissues[44] and various mutations in nAChRs that confer some resistance to 3FTXs[45] suggest this possibility.

### 3FTXs evolved explosively in snakes following the loss of a MaD region

Over a period of about 25–38 million years[46,47] *LY6*, a previously stable cluster of genes, erupted into a massive array of specialized proteins with a novel, exophysiological function: neurotoxic subjugation of prey organisms. An initial small, but significant change in a membrane-bound protein in lizards, visible both in terms of embeddings and in structure- and sequence-based trees, was followed by a removal of its membrane-anchoring region. This removal may have resulted from partial recombination with secreted members of the *LY6* cluster located farther away on the genome, i.e., *SLURP1/LYPD2*, or through a duplication of the exon that encoded the C-terminus of the protein followed by a mutation that inserted the stop codon before the MaD (a scenario that seems more likely given the previously described observations in both human *LY6K* and remnants of an anciently duplicated exon 3 in *3FTXs*–see Fig. 7). The novel secretory protein, perhaps constitutively expressed in oral glands of early snakes[48,49] may have facilitated chemical subversion of neurotransmission in prey. If, as some have conjectured[48,49], an incipient venom system has been one key adaptation behind the explosive radiation of snakes, 3FTXs may have served as one of the first major toxin families. Viperids, which do not express appreciable quantities of 3FTXs in their venoms, except for Fea's viper (*Azemiops feae*)[50], have "fully functional", but dormant, *3FTX* genes (Fig. 3). The reasons for this extreme conservation remain obscure but could include regulatory spillover from molecular mechanisms responsible for conservation of *LY6* genes, or could indicate an endophysiological role for 3FTXs in viperids, at least some of which possess 3FTX-resistant nAChRs[45].

A recent analysis of the genome of the many-banded krait (*Bungarus multicinctus*) concluded that 3FTXs may have originated via neofunctionalization of "LY6E"[41]. Our results, garnered utilizing a broader range of methods and genomic data, support the identification of *LY6E* as a close *mammalian* relative of *3FTXs*. However, we identified a group of genes that are quite likely direct descendants of transitional forms between *LY6E* and *3FTXs*. The initial expansion of *LY6E-like* genes was the result of multiplication, or mutation followed by recombination throughout the cluster, of ancestral *GPIHBP1-like* genes, the only members of the gene family present in the clawed frog (*Xenopus*) genome. *PSCA* and other a*mniote* (branch of tetrapods containing reptiles and mammals) genes (including *SLURP1/LYPD2* and *LY6H*) are also likely derived from an ancestral *GPIHBP1-like* pool of genes. Seven of nine frog genes display an alpha-helix membrane-anchoring loop, indicating the ancestrality of this feature, which is shared by almost all family members except for those in the sub-families of *SLURP1/LYPD2* and *3FTX* genes. This distinction between non-secretory, membrane bound proteins and secretory forms lacking the MaD also represents the primary disjunction, or "leap" in protein configuration space (Fig. 5). The loss of the MaD is thus relatively common in the evolution of the *LY6* family, having occurred convergently on multiple occasions, including in the origin of *3FTXs* from "pre-3FTXs".

### Snake toxin genes suggest convergent evolution

The evolutionary history of 3FTXs is marked by many unique and intriguing details. Nevertheless, its contours resemble scenarios previously described for snake venom serine proteases (SVSP) and viperid venom phospholipases A$_2$ (PLA$_2$)[1,21,22]. A more-or-less conserved group of physiologically important genes from one protein family cluster on

a chromosome with a genomic architecture shared with representatives of extant clades of tetrapods. Among mammals and toxicoferan reptiles, a particular clade of genes is exapted for new functions, usually immune functions in mammals, and toxicity in snakes. However, the change always includes toxicoferan "lizards". In each of these cases, a single gene that mutated into a form distinct from the rest of the cluster has founded a whole sub-family of genes and functions, with its members being so numerous and clinically relevant that research efforts directed at them overshadow the older and supposedly more important parental clade.

As in the pattern described for PLAs$_2$[22] and SVSPs[21], the major "neofunctionalization" events in the LY6 family follow an apparently "arbitrary" change in molecular ecology. The major structural innovation within the family concerns the loss of the MaD, i.e., a shift from a non-secretory to a secretory role. According to the classic neo-Darwinian view, such mutations are "random", or more "arbitrary". As the mutation effects are highly non-random and context-specific[1,20,51], they affect the context or molecular ecology, in which the secretory form is exposed to a novel milieu and the opportunity to engage novel interaction partners. The interactions of a secretory protein are distinct from those of a membrane-bound protein. Exposure to novel biochemical environments facilitates novel relations that originate in chance encounters between molecules and may subsequently be stabilized by selection[1]. "Recruitment" of an endophysiological protein, such as a secretory LY6, as a venom toxin with an exophysiological target, follows a further arbitrary change in molecular ecology, likely facilitated by stochastic gene expression. Thus, the transition from membrane-bound "pre-3FTXs" to functional 3FTXs likely involved a two-stage process of transitioning between molecular ecologies, from non-secretory to secretory and from endophysiological to exophysiological. However, the possibility of 3FTXs moonlighting in endophysiological roles is intriguing, and further complicates linear conceptions of neofunctionalization. Investigation of these scenarios will be further illuminated by additional high-quality caenophidian snake genomes.

We benefited from recent advances in machine learning (ML) and artificial intelligence (AI), including reliable predictions of protein 3D structure by AlphaFold2[14] and embeddings from protein Language Models (pLMs, here ProtT5[18]), to complement traditional phylogenetic and genomic tools for analysis of the 3FTX/LY6 family (Fig. 1, Table 1). We unveiled the complex evolutionary history of a fascinating and diverse gene family (Fig. 6). Paired with manual synteny analyses (Figs. 2 and 6), this approach provides an unprecedented window on protein evolution, enabling us to demonstrate that the major division within the LY6/UPAR family is not phylogenetic, but structural, resulting from loss of the membrane-anchoring region/domain (MaD, Fig. 7), which has apparently occurred multiple times in evolution. Most significantly, we could chart the evolutionary trajectory of 3FTXs from an ancestral LY6 via an intermediate, toxicoferan, membrane-bound "pre-3FTX" (Fig. 6). The functionally diverse clade of snake venom 3FTXs emerges following another independent loss of the MaD. Thus, contrary to prior hypotheses, snake venom 3FTXs cannot be said to have descended directly from either LYNXs, LY6Es or SLURPs.

## Methods

### Data genomes
All data sets were downloaded from NCBI, with exception of *Bungarus*, which is available at China National GeneBank DataBase (see accession numbers in Table 2).

### Genomic and synteny analysis
As in previous studies, including one on *3FTXs*, we used publicly available vertebrate genomes of good quality to establish locations and synteny of the *LY6* cluster. We used genomes for which verified RNA-seq genomic annotations were available as reference points and

created an extensive map of genes that populate *LY6/3FTX* clusters in those genomes. These include *TOP1MT*, *THEM6*, *cytP450*, *KCNMB2*, and others (Supplementary Fig. 1). Thereby we could establish syntenic relationships of those regions in various genomes. We then used those flanking genes as a database to BLAST[52] (NCBI-BLAST v.2.7.1+ suite, blastn, e-value cutoff of 0.05, default restrictions on word count and gaps) the genomes that were less well annotated. That gave us several hundred genomic scaffolds potentially containing *LY6* genes. We used those scaffolds for a second round of BLAST (tblastx, e-value cutoff of 0.01) against a database of sequences from well-annotated mammalian and reptilian *TOLIP* genes. Positive hits were checked visually in Geneious[53], and complete exons were manually annotated and later merged into coding sequences of newly annotated genes if the exon order and count accorded with existing reliable *LY6* annotations. We included all resulting genes that produced viable mature peptides, and then used these for the phylogenetic analysis.

## Dataset assembly
We extracted all complete gene coding regions from the genomic regions surveyed, combined them with previously compiled 3FTX datasets[11,12,41,54], and then supplemented them with all non-redundant reviewed 3FTX sequences from UniProt[55]. Duplicated, fragmented and incomplete sequences were removed. For broad comparison, we used all sequences in the TOLIP protein family in InterPro[56]. We removed signal peptides from all sequences, following SwissProt[55] annotations from humans and snakes as a guide and SignalP6 for confirmation[57].

## Phylogenetic analysis
We translated all viable genes located in the previous step into proteins and aligned these with selected publicly available sequences of interest using the L-INS-i method of MAFFT software v7.305[25] with 1000 iterations (--localpair --maxiterate 1000). These parameters were used for all subsequent alignments.

We established a naming convention to differentiate between genomic sequences (first two letters of both generic and specific epithets, followed by a number to differentiate sequences from the same scaffold). We used sequences from previously published studies and homologs from UniProt and SwissProt to provide outgroups and fill gaps in sequence space keeping UniProt or SwissProt IDs. We visually inspected the alignments for apparent errors (e.g., proper alignment of the cysteine backbone), and used to construct phylogenetic trees in ExaBayes[26] (10 parallel runs of 4 chains each, and 33,000,000 generations—which took almost 10 weeks to converge; Supplementary Fig. 2c) and iqTree[27] (100 rounds of bootstrap; Supplementary Fig. 2b).

## Embedding space analysis
Over the last few years, algorithms in natural language processing (NLP) have advanced substantially, yielding breakthroughs in tasks such as automated translation or question answering. This was achieved in particular by training Language Models (LMs) on large, but unlabeled text corpora[58,59]. Thanks to the sequential nature of both natural language and protein sequences, these advances have been transferred readily to protein sequences through so-called protein Language Models (pLMs)[17]. In both cases, masked or missing tokens (words in NLP, amino acids in computational biology) are predicted from the remaining, uncorrupted sequence context (a sentence or a protein sequence, respectively). As this objective does not require labeled data, it allows mining the wealth of unlabeled data[17–19,60], i.e., data without experimental annotations such as exponentially growing protein sequence databases by relying solely on sequential patterns in the input. Processing the information learned by such pLMs, e.g., by feeding protein sequences as inputs to the network and constructing vectors from activation in the network's last layers, yields a representation of protein sequences known as embeddings (for sketch: Fig. 1 in Elnaggar et al.[18]). This allows transfer of features learned by the

pLM to any predictive task requiring numerical protein representations (transfer learning), which has already been shown for various applications ranging from protein structure[61] to protein function[62]. Distance in embedding space correlates with protein function and can be used as an orthogonal signal for clustering proteins into functional families[62].

In this work, we generated embeddings for each protein sequence using the pLM ProtT5-XL-UniRef50[18] (for simplicity referred to as *ProtT5*), which has been built in analogy to the NLP model T5[59]. ProtT5 was trained solely on unlabeled protein sequences from BFD (Big Fantastic Database; 2.5 billion sequences including meta-genomic sequences)[63] and UniRef50. Ultimately this allowed ProtT5 to learn some of the constraints of protein sequences. As ProtT5 was only trained on unlabeled protein sequences and no supervised training or fine-tuning was performed, there is no risk of information leakage or overfitting to a certain class or label. In order to transfer the knowledge or constraints that ProtT5 had acquired to other tasks (transfer learning), we first created individual vector representations for each residue in a protein. In order to derive fixed-length vector representations for single proteins irrespective of protein length (per-protein embedding), we then averaged all residue embeddings in a protein (Fig. 1 in Elnaggar et al.[18]). As a result, every protein was represented as a 1024-dimensional embedding. Those high-dimensional representations were projected to 3D using UMAP[30] (n_neighbors = 25, min_dist = 0.5, random_state = 42, n_components = 3) and colored according to features of interest to allow visual analysis. Embeddings were created using the bio_embeddings package[64] and 2D, 3D and interactive plots using RostSpace (https://github.com/Rostlab/RostSpace).

## Protein 3D structures
For fast protein structure prediction, 3D structures for all proteins in our set were generated using default parameters of the ColabFold implementation (v1.3.0, commit: 7ebcbe62e8d88400b0e75aa0878d-ce2ff3a6c71f) of AlphaFold 2 (AF2), i.e., no early-stopping, no templates, and no amber-relaxation were used[24,65]. Input MSAs were generated using the search-script provided by ColabFold with the highest sensitivity. As AlphaFold2 is an ensemble of five models each outputting its own 3D structure, only the best 3D structure for each protein was used for further processing. For this filtering, we used the structure of the model with the highest predicted reliability, i.e., the highest predicted local distance difference test—pLDDT.

## Comparison between existing experimental 3D structures of LY6s/3FTXs and structures predicted in this study
The dataset encompassed 59 available experimental 3D structures, which were juxtaposed with the structures predicted by Alphafold2, employing Foldseek for the comparison[16]. The resultant high LDDT (0.85 ± 0.08), TM-score (0.87 ± 0.08), and RMSD (1.92 ± 0.85) provide robust evidence supporting the precision of the predicted 3D structures across the entire dataset, please see the results in Supplementary Data 1.

## Cluster analysis
Results of the embedding space analysis were used as coordinates for clustering analyses carried out in R v3.6.2[66] using RStudio 2023.03.0[67]. The most basic approach was a *k*-means clustering approach with k = 7 implemented in the default stats package[66]. We also used the DBSCAN algorithm with ε = 0.55 and minimum points = 5 to perform density-based clustering. Finally, we extracted a hierarchical density-based dendrogram from the results of the OPTICS algorithm with Ξ = 0.01. Density-based analyses were implemented in the dbscan package[68].

## 3D structure-based trees
3D structures predicted by AlphaFold2 were used as input for a local version of DALI v5 (distance matrix alignment)[28] to generate trees

based on structural similarity between all proteins in our set. All parameters were left at default, but computation was executed in parallel on 20 cores using the parameter, mpirun.

## Protein property predictions

Besides protein structures, we used various other predicted protein properties to enrich our analysis. More specifically, we used a recently published pipeline that combines a variety of protein property predictors, all of which rely on ProtT5 to encode protein sequences[69]. Here, we mainly focused on predicting whether a protein is secreted using a method called LightAttention and on predicting membrane-anchoring domains using the method, TMbed[70].

## Reporting summary

Further information on research design is available in the Nature Portfolio Reporting Summary linked to this article.

## Data availability

All data used in this study (including newly discovered sequences, embeddings and computed structures) have been deposited in the Zenodo database under accession code https://doi.org/10.5281/zenodo.8163802. Genomes accession numbers used in this study: GCA_000455745.1, GCA_000090745.2, CNP0002662, GCA_01523 7465.2, GCA_000002315.5, GCA_000001405.29, GCA_009733165.1, GCA_004115215.4, GCA_002099425.1, GCA_900067755.1, GCA_00152 7695.3, GCA_900518735.1, GCA_000186305.2, GCA_004798865.1, GCA_000004195.4.

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

## Acknowledgements

Thanks primarily to Tim Karl for invaluable help with hardware and software and to Inga Weise (TUM) for support with many other aspects of this work. Thanks to Matt Summerville and Wolfgang Wuster who gave permission to use their photographs to create snake icons, as well as Yung-Lun Lin (for *Bungarus*) and 王朝威 (for *Protobothrops*) whose images were used under Creative Commons Attribution 4.0 International license. Also thanks to all those who maintain public databases in particular Steven Burley (PDB, Rutgers), Alan Bridge (Swiss-Prot, SIB, Lausanne), Alex Bateman (UniProt, EBI Hinxton) and their crews, and to all experimentalists who enabled this analysis by making their data publicly available. IK is funded by an Alexander von Humboldt Foundation Fellowship. We also thank the Bavarian Ministry of Education for funding to the TUM through the German Ministry for Research and Education: BMBF [SSTDBB 5091431 and program 'Software Campus 2.0 (TUM) 2.0' 01IS17049]; Deutsche Forschungsgemeinschaft [DFG-GZ: RO1320/4-1].

## Author contributions

I.K., S.A., T.N.W.J. and B.R. conceived the study. I.K., T.S., D.D. and M.H. performed the experiments. All authors contributed to writing the manuscript.

## Funding

## Competing interests

The authors declare no competing interests.
