## [Peer Review File · Nature Communications]

Domain loss enabled evolution of novel functions in the snake three-finger toxin gene superfamilyREVIEWER COMMENTS

Reviewer #1 (Remarks to the Author):

This manuscript reports the compelling reconstruction of major features along the evolutionary history of the 3FTX superfamily of elapid snakes. Using a combination of phylogenetic approaches, manual synteny analysis, and cutting-edge machine learning techniques, such as AlphaFold2 to predict protein 3D structure for all 991 Ly6 family proteins and protein Language Models to generate multidimensional vector representations from single protein sequences, the authors identified the immediate ancestor of snake venom 3FTXs as a non-secretory membrane-bound Ly6, unique to squamate reptiles. The analysis indicates that i) 3FTXs evolved from a single-copy gene unique to Toxicofera reptiles, and that the loss of transmembrane region of the precursor membrane-bound "pre-3FTX2 ancestor, along with changes in gene expression, paved the way for the evolution of this important families of snake toxins. Despite the enormous amount of data analyzed, this molecular detective story is told in a rigorous but at the same time entertaining way. I have no criticism to this enormous and relevant work; just a suggestion that the authors may also find interesting to analyze:

- 1.- Is there any indication of the occurrence of exon shuffling in the translation of the genes encoding 3FTXs?
- 2.- Given that "the major structural innovation within the 3FTX family concerns the loss of the membrane region and the acquisition of a signal peptide, it would be relevant to add a figure displaying a cartoon of a proposed mechanism for this shift from a non secretory to a secretory role.
- 3.- It would be also interesting to show a cladogram of the Indian cobra 3FTXs and discuss the correlation between sequence-based molecular phylogeny and a genomic-based model for the cronological expansion of the 3FTx gene cluster.

Reviewer #2 (Remarks to the Author):

Koludarov, Heinzinger et al describe how utilizing cutting-edge methodologies in machine learning can help in elucidating phylogenetic and evolutionary questions. In this manuscript the case study is the evolution of three-finger toxins (3FTXs), deadly proteins from the venom of snakes that are distantly related to other members of the Ly6/uPAR family that in mammals was shown to play important roles in immunology and cancer. While it was clear for more than two decades that 3FTXs are related to the Ly6/uPAR family, the exact phylogenetic relationship remained elusive due to extraordinary sequence variation. The new tools employed by the authors as well as the newly available long-read genomic data from several venomous snakes help in finally revealing the evolution of this important toxin family. While the paper definitely deals with an interesting topic and demonstrates the power of novel tools, it is written in a sub-optimal format that partially masks its qualities. I believe that it will require a substantial rewriting as well as some additional analyses before its merits can be fairly judged. I detail my concerns and suggestions below:

Major comments:

1. The results of this paper are described essentially in two pages. While much attention is given to many other details throughout the paper (e.g., see below my comments about the introduction) the results are described very minimally. More details would be helpful. More specifically, it seems that the major novelty here is the use of ProtT5 protein language model (pLM). Thus, it is essential that this tool is better described. Right now, it remains more or less a "black box" to the readers.
2. The authors use more traditional phylogenetic approaches such as Bayesian Inference and Maximum Likelihood (ML; which its results are not properly supplied to the reader. See more on this below) as well as Distance-matrix ALignment (DALI). The authors show that these methods provide inconclusive results (polytomy) and even contradict one another. This is where embedding map by ProtT5 "comes to the rescue". However, the authors should also test classic pairwise comparison and clustering methods such as CLuster ANALysis of Sequences (CLANS; Frickey and Lupas 2004 *Bioinformatics* 20:3702-4). Maybe those simple methods can provide good results when compared to the embedding map? This is very important in order to evaluate how essential this novel tool may be.

3. The introduction of this manuscript is organized in a very unusual way. While styling should be flexible and each researcher might have their own preferences when it comes to scientific writing, here the structure of the introduction hinders the ability of the reader to understand the essential background for this study. The introduction is very comprehensive and is even longer than the results. It is currently structured as a mini-review. It has sections with their own titles and within this unorthodox structure when the authors refer to their own novel results from the current paper (Page 4 lines 43-46) the reader can get confused. Altogether, I would strongly recommend that the authors try to re-write their introduction in a slightly more succinct way (include only the essentials) and preferably get rid of the headings for the different sub-sections and try to connect them into a clearer narrative.

4. Many of the figures are unintuitive and require either more detailed figure legends or better graphical keys in the figure itself, or even both. Just an example: in Figure 4A, what do the different silhouettes/cartoons of the three snake lineages mean? I guess the one with the "hood" is a cobra? So these would be Elapids? So one of the others without the "hood" is Viperids? What is the third snake lineage? And most importantly, why do I need to guess?

5. Page 12 lines 33-34 states: "A recent analysis of the genome of *Bungarus multicinctus* (published too late for inclusion in our database)". Honestly, I never encountered such a statement in a scientific paper. Now, one could imagine that this paper came out two days before the current manuscript was submitted and the authors were forced to include it at the very last moment (and even then this statement would be irrelevant). However, this paper was actually published in July 2022 (Zhang et al. 2022 Cell Reports 40: 111079). With no disrespect intended, we are talking here about constructing phylogenies and computational analyses, not an experiment requiring years of lab work. Hence, I believe the analyses should be re-run with the *Bungarus* data and its genomic structure should be included in the figures. This is a very valuable dataset as it is a chromosome-level assembly of a krait, which is an elapid. A key lineage for this analysis. The authors even state in Page 12 lines 5-6 "Accumulation of 3FTx genes in the Indian cobra (*N. naja*) genome (the only elapid genome available on the chromosome-level) provides a vivid example of this process.". I understand there is not much joy in re-analyzing everything again with the *Bungarus* data, but it is essential in order to see whether the main conclusions hold true when it is included.

Minor comments:

1. The structure of the title is very confusing. I would suggest either to shorten it to "Domain loss enables evolution of novel functions in a gene superfamily" or change it to "Domain loss enables evolution of novel functions in 3-Finger Toxins of the Ly6/uPAR gene superfamily" and by the way I believe Ly6/uPAR is usually considered as a gene family and not a superfamily, so please consider changing this as well throughout the text.
2. Abstract: name the "cutting-edge machine learning techniques".
3. Page 2 line 23: I suggest changing "derivatives" to "members".
4. Page 4 line 16: delete "non-specific".
5. Page 5 lines 2-3 No doubt that AlphaFold2 made a revolution in biology. Yet, no need to declare in the introduction of a research article that it is "Nature's Method of the Year 2021".
6. Page 5 line 10: "aspects of the language of life as written in protein sequences" is very prosaic. Please rephrase.
7. Page 9 line 10: you already mentioned before what pLM stands for so no need to do it again (Page 4). In general, the definition of pLM is mentioned several times in the text. Please remove this redundancy.
8. For some reason in the PDF the references are very messy and are not aligned. Please fix.
9. In the supplementary figures some 3FTxs are referred to as "weird". I suggest to reconsider this term. Maybe "unusual" is more fitting? In any case it is essential to explain why they got this definition.
10. The authors provide the ML and DALI phylogenies in Newick format. This is insufficient as some readers of Nature Communications have zero experience with phylogeny software and would not be able to visualize the trees. Hence, it is crucial that the authors also include PDF (or other graphic) versions of the trees in the supplementary data and color and name the nodes in identical manner to the Bayesian phylogeny so even an inexperienced reader can easily compare the phylogenetic trees.

RESPONSE TO REVIEWERS' COMMENTS

Comments Reviewer #1

Thank you for your positive feedback on our manuscript. We appreciate your recognition of our study, and we look forward to sharing our research with others in the field.

Q This manuscript reports the compelling reconstruction of major features along the evolutionary

history of the 3FTX superfamily of elapid snakes. Using a combination of phylogenetic approaches, manual synteny analysis, and cutting-edge machine learning techniques, such as AlphaFold2 to predict protein 3D structure for all 991 Ly6 family proteins and protein Language Models to generate multidimensional vector representations from single protein sequences, the authors identified the immediate ancestor of snake venom 3FTXs as a non-secretory membrane-bound Ly6, unique to squamate reptiles. The analysis indicate that i) 3FTXs evolved from a single-copy gene unique to Toxicofera reptiles, and that the loss of transmembrane region of the precursor membrane-bound "pre-3FTX2 ancestor, along with changes in gene expression, paved the way for the evolution of this important families of snake toxins. Despite the enormous amount of data analyzed, this molecular detective story is told in a rigorous but at the same time entertaining way. I have no criticism to this enormous and relevant work; just a suggestion that the authors may also find interesting to analyze:

1.-. Is there any indication of the occurrence of exon shuffling in the translation of the genes encoding 3FTXs?

A As far as we know there is no indication of exon shuffling in 3FTx subfamily. All duplicated exons have stop codons inserted, rendering them inactive. Various orphan exons are located too far from other genes to be considered anything, but remnants of old duplication events. The few instances of exon shuffling we uncovered were restricted to Ly6 genes – the SLURP-LYPD readthrough.

Q 2.- Given that "the major structural innovation within the 3FTX family concerns the loss of the membrane region and the acquisition of a signal peptide, it would be relevant to add a figure displaying a cartoon of a proposed mechanism for this shift from a non secretory to a secretory role.

A This is a valid point, and we fully agree. To address this suggestion, we added Figure 7, which includes a schematic explaining how both human Ly6 and snake forms may have evolved from "generic" Ly6. We also expanded Figure 1 to include a schematic breakdown of Ly6/3FTx molecules and how they fit into the exonic structure of a gene, so that Figure 7 has more context.

Q 3.- It would be also interesting to show a cladogram of the Indian cobra 3FTXs and discuss the correlation between sequence-based molecular phylogeny and a genomic-based model for the cronological expansion of the 3FTx gene cluster.

A To address this matter, we added Figure 3 panel B; however, the relationships between the numerous cobra genes are convoluted and without a comprehensive comparative genomic study, impossible to properly disentangle at the moment, due to lack of contiguous elapid genomes in desired numbers. It is infeasible to propose genomic mechanisms that created the cobra cluster, though some gene duplications became apparent on the added Figure 3 panel B.

Comments Reviewer #2

Thank you for your thoughtful comments on our manuscript. We appreciate your recognition of our utilization of cutting-edge methodologies to resolve phylogenetic and evolutionary questions. We have addressed your criticisms by substantially changing the paper and including additional analyses. We hope that the revisions we have made have clarified the strengths of our study, and we look forward to hearing your updated feedback. Thank you again for your time and valuable input.

Q Koludarov, Heinzinger et al describe how utilizing cutting-edge methodologies in machine learning can help in elucidating phylogenetic and evolutionary questions. In this manuscript the case study is the evolution of three-finger toxins (3FTxs), deadly proteins from the venom of snakes that are distantly related to other members of the Ly6/uPAR family that in mammals was shown to play important roles in immunology and cancer. While it was clear for more than two decades that 3FTxs are related to the Ly6/uPAR family, the exact phylogenetic relationship remained elusive due to extraordinary sequence variation. The new tools employed by the authors as well as the newly available long-read genomic data from several venomous snakes help in finally revealing the evolution of this important toxin family. While the paper definitely deals with an interesting topic and demonstrates the power of novel tools, it is written in a sub-optimal format that partially masks its qualities. I believe that it will require a substantial rewriting as well as some additional analyses before its merits can be fairly judged. I detail my concerns and suggestions below:

Major comments:

1. The results of this paper are described essentially in two pages. While much attention is given to many other details throughout the paper (e.g., see below my comments about the introduction) the results are described very minimally. More details would be helpful. More specifically, it seems that the major novelty here is the use of ProtT5 protein language model (pLM). Thus, it is essential that this tool is better described. Right now, it remains more or less a “black box” to the readers.

A We have expanded the Results and Discussion sections, in particular lines 300-325. We also revised the introductory section on protein language models. It is worth noting, however, that despite all attempts to understand the workings of AI, there is still a lot that science does not understand about it.

Q 2. The authors use more traditional phylogenetic approaches such as Bayesian Inference and Maximum Likelihood (ML; which its results are not properly supplied to the reader. See more on this below) as well as Distance-matrix ALIGNment (DALI). The authors show that these methods provide inconclusive results (polytomy) and even contradict one another. This is where embedding map by ProtT5 “comes to the rescue”. However, the authors should also test classic pairwise comparison and clustering methods such as CLuster ANalysis of Sequences (CLANS; Frickey and Lupas 2004 *Bioinformatics* 20:3702-4). Maybe those simple methods can provide good results when compared to the embedding map? This is very important in order to evaluate how essential this novel tool may be.

A The polytomy of even the best phylogenetic trees of this protein family was one of the reasons that we turned to protein language models. We fully agree that using simpler methods is better than using complicated ones, if the results are the same. To check that, we added CLANS analysis to the study (now part of the Supplementary Material). In brief, the clustering is comparable to that of pLMs; however, it is noisier and less comprehensive, with multiple data points positioned far from any cluster. We would like, however, to point out that CLANS

Response to Reviewers on “Domain loss enabled evolution of novel functions in a gene superfamily”

is a 2-dimensional analysis, while ours has 1024 dimensions, and even when collapsed into 3D, it adds a lot. In addition, pLM representations allow for many downstream tasks, one of which – membrane localisation prediction – was a crucial part of this study.

Q 3. The introduction of this manuscript is organized in a very unusual way. While styling should be flexible and each researcher might have their own preferences when it comes to scientific writing, here the structure of the introduction hinders the ability of the reader to understand the essential background for this study. The introduction is very comprehensive and is even longer than the results. It is currently structured as a mini-review. It has sections with their own titles and within this unorthodox structure when the authors refer to their own novel results from the current paper (Page 4 lines 43-46) the reader can get confused. Altogether, I would strongly recommend that the authors try to re-write their introduction in a slightly more succinct way (include only the essentials) and preferably get rid of the headings for the different sub-sections and try to connect them into a clearer narrative.

A We shortened the introduction and added references to the existing reviews as necessary, in case the reader wants to know more. We removed the headings, although we strongly felt that they decrease confusion and allow the reader to skip parts they don't understand and to quickly find parts they want to read. We agree that this is an unorthodox method of achieving that. Instead, we streamlined the narrative and we hope it now reads significantly better.

Q 4. Many of the figures are unintuitive and require either more detailed figure legends or better graphical keys in the figure itself, or even both. Just an example: in Figure 4A, what do the different silhouettes/cartoons of the three snake lineages mean? I guess the one with the “hood” is a cobra? So these would be Elapids? So one of the others without the “hood” is Viperids? What is the third snake lineage? And most importantly, why do I need to guess?

A We fully agree that this was a flaw in the original manuscript and have added a panel to Figure 1 to provide a clear connection between the icons used and species names. In addition, where differentiating between the snake species gets critical (Figure 3, panel A), we used color icons to make each of the snakes stand out as well as clarifying the species in the figure legend. In addition, we added explanatory figure legends to the majority of the figures and doubled down on our efforts to make them clear. It is worth noting, that the topic at hand is very complex and it is not a trivial task to clearly present gene-genomic relationships on this scale. We hope, however, that our efforts were worthwhile.

Q 5. Page 12 lines 33-34 states: “A recent analysis of the genome of Bungarus multicinctus (published too late for inclusion in our database)”. Honestly, I never encountered such a statement in a scientific paper. Now, one could imagine that this paper came out two days before the current manuscript was submitted and the authors were forced to include it at the very last moment (and even then this statement would be irrelevant). However, this paper was actually published in July 2022 (Zhang et al. 2022 Cell Reports 40: 111079). With no disrespect intended, we are talking here about constructing phylogenies and computational analyses, not an experiment requiring years of lab work. Hence, I believe the analyses should be re-run with the Bungarus data and its genomic structure should be included in the figures. This is a very valuable dataset as it is a chromosome-level assembly of a krait, which is an elapid. A key lineage for this analysis. The

Response to Reviewers on “Domain loss enabled evolution of novel functions in a gene superfamily”

authors even state in Page 12 lines 5-6 “Accumulation of 3FTx genes in the Indian cobra (*N. naja*) genome (the only elapid genome available on the chromosome-level) provides a vivid example of this process.”. I understand there is not much joy in re-analyzing everything again with the *Bungarus* data, but it is essential in order to see whether the main conclusions hold true when it is included.

A We fully agreed that adding *Bungarus* genomic information to the paper would make results more robust, thus we added all sequences from that study to our dataset and annotated those that were missed by the original study (some were later removed due to being duplicates of other sequences).

We added those sequences and the genomic organisation of the *Bungarus* 3FTx cluster to all of our analyses and re-ran them. However, we decided not to clutter figures that deal with the general layout of the cluster because of the complexity of *Bungarus* and its similarity to *Naja*. Figure 3, panel A now has *Bungarus* genomic organisation along with other snakes. However, the *Bungarus* genome is not quite chromosomal-level, despite being extremely contiguous. Twelve of the identified 3FTxs are located on several UN-placed scaffolds; thus, direct synteny comparison is possible only for the chromosomal part.

Q Minor comments:

1. The structure of the title is very confusing. I would suggest either to shorten it to “Domain loss enables evolution of novel functions in a gene superfamily” or change it to “Domain loss enables evolution of novel functions in 3-Finger Toxins of the Ly6/uPAR gene superfamily” and by the way I believe Ly6/uPAR is usually considered as a gene family and not a superfamily, so please consider changing this as well throughout the text.

A We modified the title accordingly, however “superfamily” has been retained, because “superfamilies are defined as a group of proteins or genes of common origin with nonoverlapping functions” (Ohta, 2008 <https://doi.org/10.1002/9780470015902.a0005126.pub2>) and “a large group of distantly related proteins” (<https://www.ebi.ac.uk/training/online/courses/protein-classification-intro-ebi-resources/protein-classification/what-are-protein-families/>)

Q 2. Abstract: name the “cutting-edge machine learning techniques”.

A We added the following to the sentence: “including AlphaFold and ProtT5”

Q 3. Page 2 line 23: I suggest changing “derivatives” to “members”.

A Changed to “derived members” (that they are derived, rather than plesiotypic, a key aspect of the study).

Q 4. Page 4 line 16: delete “non-specific”.

A This section of the Introduction has been deleted.

Q 5. Page 5 lines 2-3 No doubt that AlphaFold2 made a revolution in biology. Yet, no need to declare in the introduction of a research article that it is “Nature’s Method of the Year 2021”.

A Fair point. This was removed.

Response to Reviewers on “Domain loss enabled evolution of novel functions in a gene superfamily”

Q 6. Page 5 line 10: “aspects of the language of life as written in protein sequences” is very prosaic. Please rephrase.

A Rephrased to “Thereby protein Language Models (pLMs) learn to understand some aspects of the “language” of protein sequences.”

Q 7. Page 9 line 10: you already mentioned before what pLM stands for so no need to do it again (Page 4). In general, the definition of pLM is mentioned several times in the text. Please remove this redundancy.

A Redundancies removed.

Q 8. For some reason in the PDF the references are very messy and are not aligned. Please fix.

A Done.

Q 9. In the supplementary figures some 3FTxs are referred to as “weird”. I suggest to reconsider this term. Maybe “unusual” is more fitting? In any case it is essential to explain why they got this definition.

A Thank you for pointing this out. We changed it to “Non-standard” and expanded the text as well amended the figures to clarify that we refer to sequences that do not fall into other established categories (plesiotypic, short-chain and long-chain). We added a point that other studies that use similar terminology (“non-canonical” in Zhang et al. 2022).

Q 10. The authors provide the ML and DALI phylogenies in Newick format. This is insufficient as some readers of Nature Communications have zero experience with phylogeny software and would not be able to visualize the trees. Hence, it is crucial that the authors also include PDF (or other graphic) versions of the trees in the supplementary data and color and name the nodes in identical manner to the Bayesian phylogeny so even an inexperienced reader can easily compare the phylogenetic trees.

A Circular trees for easy visual comparison of branching have been added to the main text as Figure 4. Horizontal full trees for detailed inspection have been added to the Supplementary Material as graphic figures. However, the addition of *Bungarus* and some other genes flagged by experts in the field that contacted us independently have resulted in ML phylogeny resembling Bayesian phylogeny more than DALI. DALI and Bayesian phylogeny remained largely unchanged.

REVIEWERS' COMMENTS

Reviewer #1 (Remarks to the Author):

Authors have addressed all my comments and I'm more than happy to endorse publication of their seminal article that solves the riddle of the evolutionary origin of one of the most, if not the most, important toxin superfamilies of Elapidae venoms. Congrats!

Reviewer #2 (Remarks to the Author):

The authors have adequately addressed all my comments from the previous round. I do not have any further concerns.